# Chemical Composition, In Vivo, and In Silico Molecular Docking Studies of the Effect of *Syzygium aromaticum* (Clove) Essential Oil on Ochratoxin A-Induced Acute Neurotoxicity

**DOI:** 10.3390/plants14010130

**Published:** 2025-01-04

**Authors:** Mostapha Brahmi, Djallal Eddine H. Adli, Imane Kaoudj, Faisal K. Alkholifi, Wafaa Arabi, Soumia Sohbi, Kaddour Ziani, Khaled Kahloula, Miloud Slimani, Sherouk Hussein Sweilam

**Affiliations:** 1Department of Biological Science, Faculty of Natural and Life Sciences, University of Ahmed Zabana, Relizane 48000, Algeria; 2Laboratory of Biotoxicology, Pharmacognosy and Biological Valorization of Plants (LBPVBP), Department of Biology, Faculty of Sciences, University of Dr MoulayTahar, Saida 20000, Algeria; djillou2006@yahoo.fr (D.E.H.A.); imane.kaoudj20@gmail.com (I.K.); wafaaarabi@yahoo.fr (W.A.); s.soumiabio@yahoo.fr (S.S.); zianivet07@gmail.com (K.Z.); bombixc2@yahoo.fr (K.K.); mslimani20@gmail.com (M.S.); 3Department of Pharmacology, College of Pharmacy, Prince Sattam Bin Abdulaziz University, Al-Kharj 11942, Saudi Arabia; f.alkholifi@psau.edu.sa; 4Department of Pharmacognosy, College of Pharmacy, Prince Sattam Bin Abdulaziz University, Al-Kharj 11942, Saudi Arabia; 5Department of Pharmacognosy, Faculty of Pharmacy, Egyptian Russian University, Badr City, Cairo-Suez Road, Cairo 11829, Egypt

**Keywords:** ochratoxin A, *Syzygium aromaticum*, GC/MS, neurobehavioral tests, biochemical parameters, in silico

## Abstract

The aim of our research was to understand the impact of ochratoxin A (OTA) exposure on various physiological and behavioral aspects in adult Wistar rats, and to evaluate the efficacy of a *Syzygium aromaticum* essential oil (EOC) treatment in restoring the damage caused by this toxin. The essential oils were extracted by hydrodistillation, a yield of 12.70% was obtained for EOC, and the GC-MS characterization of this essential oil revealed that its principal major components are eugenol (80.95%), eugenyl acetate (10.48%), β-caryophyllene (7.21%), and α-humulene (0.87%). Acute OTA intoxication was induced by an intraperitoneal (IP) injection of 289 µg/kg/b.w. every 48 h for 12 doses, resulting in significant reductions in the body and brain weights of exposed rats when compared with controls. The neurobehavioral analysis using several behavioral testing techniques, such as the forced swimming, the dark/light test, the Morris water maze, and the open field test, clearly revealed that OTA exposure causes neurobehavioral disorders, including decreased locomotor activity, a reduced willingness to explore the environment, reflecting a state of stress, anxiety and depression, as well as impaired memory and learning. In addition, OTA intoxication has been associated with metabolic disturbances such as hyperglycemia and hypercortisolemia. However, treatment with EOC mitigated these adverse effects by improving body and brain weights and restoring neurobehavioral function. The in silico analysis revealed significant affinities between clove oils and two tested esterase enzymes (ACh and BuChE) that were more than or similar to the four neurotransmitters “dopamine, serotonin, norepinephrine, and glutamic acid” and the co-crystallized ligands NAG, MES, and GZ5. These results highlight the therapeutic potential of EOC in combating the toxic effects of OTA and pave the way for future research into the mechanisms of action and therapeutic applications of natural compounds in the prevention and treatment of poison-induced diseases.

## 1. Introduction

Ochratoxin A (OTA) is a potent mycotoxin produced by fungi of the *Aspergillus* and *Penicillium* genera that is commonly found contaminating a variety of food products, such as cereals, coffee, dried fruits, and wine. Due to its widespread occurrence and severe toxicological effects, OTA poses a significant risk to human and animal health. It is known to cause nephrotoxicity, hepatotoxicity, and neurotoxicity, with the neurotoxic effects being particularly concerning due to the potential for chronic exposure to low doses over time [1]. Mechanistically, OTA induces oxidative stress, disrupts mitochondrial function, and leads to neuronal damage, which cumulatively result in both acute and chronic neurotoxic outcomes [2].

Given the serious health implications of OTA exposure, there is a pressing need to identify effective protective agents. Natural compounds have gained attention for their potential as protective agents due to their bioactivity and lower side effects. *Syzygium aromaticum*, commonly known as clove, is one such natural product. The essential oil (EO) derived from clove is rich in eugenol, a phenolic compound renowned for its antioxidant, anti-inflammatory, and antimicrobial properties [3]. Studies have demonstrated the efficacy of the EOC in counteracting oxidative stress and protecting against various toxic agents, making it a promising candidate for mitigating OTA-induced neurotoxicity [4].

To fully understand the neuroprotective potential of the clove essential oil (EOC), it is crucial to employ both in vivo and in silico approaches. In vivo studies are invaluable for providing empirical evidence of the biological effects and therapeutic potential of compounds in living organisms. These studies can elucidate the pharmacodynamics and pharmacokinetics of EOC and its primary constituents, providing insights into its overall efficacy and safety profile. In contrast, in silico molecular docking studies offer a complementary perspective by predicting the interactions at the molecular level between its major bioactive compounds and specific biological targets. This approach can reveal the potential mechanisms of action and identify the key molecular pathways involved in the protective effects of EOC [5,6]

In this study, we aim to characterize the chemical composition of *Syzygium aromaticum* essential oil and assess its neuroprotective effect against OTA-induced acute neurotoxicity in rats. Furthermore, we employ in silico molecular docking studies to explore the potential interactions between the main constituents of EOC and relevant molecular targets implicated in OTA toxicity. This dual approach not only provides a comprehensive understanding of the protective effects of EOC but also identifies potential molecular mechanisms, thereby advancing its potential application as a therapeutic agent against OTA-induced neurotoxicity.

## 2. Results

### 2.1. Essential Oil (EOC) Yields and Main Compounds

The hydro-distillation method for Syzygium aromaticum yielded an essential oil (EOC) of 12.70%. An analysis of Syzygium aromaticum essential oil by gas chromatography identified 26 main compounds, which are presented in Table 1. These compounds represented a total of 80.93% phenols, 8.56% sesquiterpene hydrocarbons, 10.48% esters, and 0.03% terpene alcohols. The main compounds of this oil were eugenol (80.83%), eugenyl acetate (10.48%), β-caryophyllene (7.21%), and α-humulene (0.87%) (Table 1).

### 2.2. Effects of Ochratoxin A on Body Weight and Brain Weight

This study involved the determination of the body and brain weights of adult rats intoxicated with OTA compared with control rats (Table 2). The results of this study revealed a significant reduction in the body weight of OTA-intoxicated rats compared with control rats and a significant decrease in the brain weight. In contrast, the intraperitoneal administration of EOC at a dose of 0.1 mg/mL resulted in significant increases in the body and brain weights compared with intoxicated rats. However, brain weights in the group of EOC-treated control rats showed no difference compared with control rats.

### 2.3. Neurobehavioral Tests

#### 2.3.1. Forced Swimming Test

##### Immobility Time (TIM)

The forced swimming test results revealed a significantly (*p* < 0.050) longer immobility time (TIM) for OTA-intoxicated rats than for control rats. This increase explains the animal’s inability to swim due to a decrease in muscle tone and the onset of desperation behavior. In addition, the intraperitoneal administration of EOC (clove essential oil) resulted in a decrease in the TIM of intoxicated and treated (EOC) rats compared to intoxicated rats. In the same context, we observed a significantly greater decrease (*p* < 0.001) in the TIM of intoxicated rats treated with EOC (OTA-EOC) than that of intoxicated rats, which could be explained by a reduction in despair behavior (Figure 1).

#### 2.3.2. Open Field Test

This test measures the reaction of the animal to a particular new environment and their motivation to explore the space. The recorded results showed that OTA-intoxicated rats showed locomotor hypoactivity (horizontal and vertical), as represented by the number of squares crossed and the number of rightings, respectively (*p* < 0.001; *p* < 0.001), compared with control rats, and the results also showed the development of a state of anxiety in the intoxicated group (OTA), which was observed by a significant drop in the number of visits to the center compared with control rats (*p* < 0.001), as well as a significant increase in the latency time (*p* < 0.001) (Figure 2).

Additionally, the administration 0.1 mg/kg of EOC intraperitoneally to intoxicated rats resulted in a significantly greater increase (*p* < 0.001) in the locomotor activity of intoxicated (OTA) and treated rats in terms of righting and the number of tiles crossed, as well as a significant reduction (*p* < 0. 001) in the anxiety state of these rats, as assessed by the number of visits to the center, and subsequently, no significant difference was noticed in the numbers of grooming behaviors and defecations (*p* < 0.001; *p* < 0.001).

#### 2.3.3. Dark/Light Test

The results revealed that OTA-intoxicated rats spent significantly more time in lighted compartments (*p* < 0.050) than control rats, which spent more time in dark compartments, indicating anxiety behavior in OTA-intoxicated rats. In addition, the intraperitoneal administration of EOC resulted in a significant reduction (*p* < 0.050) in the time spent in the lighted compartment by OTA-EOC-treated rats compared with intoxicated rats. Non-intoxicated rats treated with EOC also showed a positive result compared with control rats (Figure 3).

#### 2.3.4. Morris Water Maze Test

As this test evaluates spatial memory and learning, including the acquisition of visual spatial data, it is an important tool for validating rodent models in the study of neurocognitive disorders. On the first test days, no differences were observed in the rodents, implying that all rodents were able to see the platform and landmark pavilion in the surrounding environment, and could swim acceptably. During the four days when the platform was hidden, differences were observed in the escape latency, path length, and latency time, showing that OTA-intoxicated rats had a significantly (*p* < 0.001) higher latency time than control rats. Additionally, the group of intoxicated rats that were treated (OTA-EOC) showed improvements in memory capacity and spatial learning, as represented by a reduction in latency, and showed a shorter latency to escape to the hidden platform on days 3 and 4, as well as a shorter swimming length before escaping to the hidden platform (Figure 4). Moreover, the probe test assessed memory capacity, with OTA-intoxicated rats spending significantly less time (*p* < 0.001) in the target quadrant (NO: where the hidden platform was previously placed) than control rats.

In addition, the injection of EOC into intoxicated (OTA) rodents resulted in a significantly greater increase (*p* < 0.001) in the memory capacity, as represented by an increase in time spent in the target quadrant (NO) compared to intoxicated rats (Figure 5).

In the same way, the results of the vision test showed that all rats, whether intoxicated, control, or treated control, were able to see the flag and then the fixed platform (Figure 6).

### 2.4. Biochemical Assays

#### 2.4.1. Blood Glucose Levels

The results showed a significant increase (*p* < 0.05) in blood glucose levels in intoxicated rats (OTA) compared with control © rats. However, treatment with EOC resulted in a significant (*p* < 0.05) decrease in blood glucose levels in treated and intoxicated (OTA-EOC) rats compared with intoxicated (OTA) rats (Figure 7).

#### 2.4.2. Cortisol Levels

The results showed a significant increase (*p* < 0.05) in cortisol levels in intoxicated (OTA) rats compared with control rats. However, the injection of EOC into intoxicated rats showed a significant decrease (*p* < 0.001) in cortisol levels compared with intoxicated (OTA) rats, demonstrating its anxiolytic effects (Figure 8).

### 2.5. Histological Study

The histological study of the brain tissue showed that OTA caused several cerebral lesions, particularly hemorrhages in different sections of the brain (Figure 9A).

Neurons in most parts of all groups of rats were perfectly normal, although in some areas, neurodegenerative changes were noted with swelling of the astrocytes and focal gliosis (Figure 9B). In the same context, in the cerebellum, we observed a very small number of eosinophilic Purkinje neurons (Figure 9C), as well as marked congestion and hemorrhages in the brain and meninges. In addition, EOC treatment had the same effect on repairing the brain tissue, showing a more or less normal tissue structure (Figure 9D,E).

### 2.6. In Silico Study of Neuroprotective Molecular Targets

The molecular docking strategy involves linking the in vitro, in vivo, and in silico studies. After docking our researched chemicals with all the explored targets for this investigation, the purpose of this study was to assess the effect of the clove essential oil on the enzymes of the cholinergic system. In the binding sites of the human acetylcholinesterase and butyrylcholinesterase receptors, the two predominant two monoterpenes (eugenol and eugenol acetate) from clove essential oil were identified. We have found binding interactions for all of them except the neurotoxic substance “ochratoxin A” with the same binding site of the human acetylcholinesterase receptor. The in-silico study of these two components, one neurotoxic, and four neuro-controls was conducted, and the results are shown in Table 3 and Figure 10 and Figure 11.

The majority of the tested components showed inhibitory activities toward these enzymes, with variations in the binding affinities from −3.2 to −6.6 Kcal/mol, and showed best fit positions or similarity to the co-crystallized ligands NAG (2–acetamido–2–deoxy-beta–D–glucopyranose), MES (2–(n-morpholino)–ethanesulfonic acid), and GZ5 ((2–{R-azanyl–{N}s–[6–[(6–chloranyl–1,2,3,4–tetrahydroacridin–9–yl)amino]hexyl]–3–(1–{H}–indol–3–yl)propenamide) for the human butyrylcholinesterase receptor and NAG only with the human acetylcholinesterase receptor. 

#### 2.6.1. Interactions with PDB ID: 4EY5 “Human Acetylcholinesterase Receptor”

Acetylcholinesterase (ACh) is a necessary neurotransmitter that plays a role in the impulses that are transmitted across synapses and toward the effector organ. After being released from the pre-synaptic nerve terminal, ACh interacts with its receptors to activate them, which then transmits a strong impulse towards the post-synaptic neuron. These esterases cause a pre-synaptic nerve terminal to stop functioning by breaking down and recycling ACh. A considerable ACh shortage results from the specific degradation of cholinergic neurons in brain illnesses, which makes it more difficult to consolidate memories and perform daily activities. One therapeutic approach to extend the duration of the ACh produced at the nerve terminal and enhance the response is to inhibit the esterases that metabolize ACh [7].

Two of the tested compounds (eugenol and its acetate derivative) and four neuro-controls displayed energy affinities from −4.9 to −6.6 Kcal/mol more than NAG (co−crystalline ligand) with −4.4 Kcal/mol, which determined the key locations for binding at the target enzyme cavity (Table 3). Eugenol acetate formed a higher fit position within the enzyme than eugenol, with a BA of −6.6 Kcal/mol, the reference ligand, and all tested neurotransmitters. Its binding affinity was established by one conventional H bond with Ser293, one carbon–hydrogen bond with Arg296, and five pi–hydrophobic bonds with Trp286 (pi–pi stacked interaction), Tyr72, Try124, and Try341 (pi–alkyl interaction) residues. Meanwhile, eugenol formed one conventional H bond with Gln291 and four pi–hydrophobic bonds with Trp286 (pi–sigma and pi–pi stacked interactions), Leu289, and Tyr72 (pi–alkyl interactions) with a BA of −5.9 Kcal/mol. It is worth noting that the neurotoxic substance ochratoxin A showed binding affinity for this enzyme (−4.2 Kcal/mol). All interactions with tested substances were displayed with many interactions (van der Waals bonds) in Figure 10 that illustrate the correlations between the number and the types of favorable bonds and the binding affinity of each component with the core amino acid residues of the target enzyme.

#### 2.6.2. Interactions with PDB ID: 6I0C “Human Butyrylcholinesterase Receptor”

All tested components established binding affinities from −3.2 to 5.2 Kcal/mol, which were more than the co-crystallized ligand (GZ5) at −2.8 Kcal/mol, and all neurotransmitter controls showed BAs more than the MES co-crystallized ligand. However, all tested compounds, eugenol and its acetate derivative, had Bas of −4.7 and −5.2 Kcal/mol, respectively. The four neurotransmitters “dopamine, serotonin, norepinephrine and glutamic acid” had BAs from −4.9 to −5.2 Kcal/mol that were less than the co-crystallized ligand NAG at −5.5 Kcal/mol, which was established in Table 3, against the tested BuChE enzyme. Concerning that, eugenol acetate formed two conventional H bonds with the Asn289 residue and one carbon–hydrogen bond with the Ser287 residue. On the other hand, eugenol established a conventional H bond with the Gln119 residue together with a pi–donor hydrogen bond with the Asn68 residue, and an alkyl bond with the Ala277 residue, together with many interactions (van der Waals bonds) in Figure 11 that illustrate the correlations between the number and the types of favorable bonds and the binding affinity of each component with the core amino acid residues of the target enzyme. It bears noting that clove oils showed BAs that were significantly higher than ochratoxin A (−3.2 Kcal/mol) for this enzyme.

## 3. Discussion

*Syzygium aromaticum* essential oil has been extracted by hydrodistillation, with a yield of 12.70%. Our results disagree with the work carried out by Adli et al., 2017 [8], who obtained a yield of 10.60%, while other research by Selles et al., 2010 [9], recorded a yield of 11.6% ± 0.7. Additionally, studies by Alfikri et al., 2020 [10], showed that the flower buds of young trees produced a higher yield (16.73%) than those of mature trees (14.93%), and that the quantity of the most abundant molecule of *Syzygium aromaticum*, which is eugenol, increased with the maturity of the flowering stages.

As reported by Brahmi et al., 2020, and Djermane et al., 2022 [11,12], the yield variation can be explained by a number of factors, such as climate, the geographical origin of the species, the harvesting period, the drying time, and the method used for the extraction of the essential oil.

An analysis of EOC by chromatography identified 26 volatile compounds, the main ones being eugenol (80.83%), eugenyl acetate (10.48%), and β-caryophyllene (7.21%). Our results are largely similar to published data by Chaieb et al., 2007 [3], who identified 24 compounds in clove essential oil, with eugenol predominating at around 78%, followed by eugenyl acetate at around 10%, and β-caryophyllene at around 7%. Similarly, the study by Singh et al., 2018 [13], reported 23 compounds in the essential oil, with a composition of 76–85% eugenol, 8–12% eugenyl acetate, and 5–8% β-caryophyllene, which also correspond well with our results. Thus, our analyses confirm and are consistent with the compositions found in the literature, demonstrating the consistency of the main constituents of *Syzygium aromaticum* essential oil across different studies.

Intraperitoneal exposure of Wistar rats to a dose of 289 µg/Kg OTA for 2 weeks caused significantly reduced weights of the body and brain compared with control rats. Our results are in accordance with Nogaim et al., 2020 [14], who observed a decrease in the body weight of rats given an oral dose of (20 mg/kg) OTA. The results are also in consistent with Izco et al., 2021 [15], who noted decreases in rat body weight and brain weight after the oral administration of 0.21, 0.5, and 1.5 mg/Kg. These were due to reduced food intake. In fact, OTA is known to inhibit protein synthesis, leading to a disruption of protein metabolism. This mechanism can be attributed to its weight-loss effect [16,17], while the decrease in brain weight can be attributed to ochratoxin A-induced damage.

Otherwise, we recorded significant increases in the body and brain weights of rats in the group treated with EOC by intraperitoneal injection compared with the OTA-treated group. These results are in agreement with Bakour et al., 2018 [18], who noted increases in the body and brain weights of their experimental rats after the administration of a *Syzygium aromaticum* extract.

This increase may be due to the corrective effects of the essential oil on ochratoxin A-induced damage, and according to Taroq et al., 2021 [19], the addition of *Syzygium aromaticum* essential oil (EOC) significantly increases food intake in broiler chicks, which can be attributed to terpenoids that act by stimulating glucose transport in cells [20].

The open field test is a widely used test for assessing locomotion and anxiety in rodents [21]. Our results indicated that the exposure of Wistar rats to OTA induced hypoactivity, which was reflected by decreases in the numbers of tiles crossed and center visits, while an increase in certain stereotyped behaviors (defecation, grooming, and righting) with a high latency was recorded, which reflect an anxious state of the intoxicated rat.

The research conducted by Tanaka et al., 2016 [22] examined the effects of OTA exposure on GABAergic interneurons, which are known for their crucial roles in neuronal development and function in the adult dentate gyrus [23]. Dysfunction of GABAergic interneurons can lead to a variety of neuropsychiatric disorders, including anxiety and depression [24], as confirmed by the study by Zhu et al., 2019 [25], showing a correlation between a decrease in these interneurons and anxious behaviors in mice. GABA is known to have an anxiolytic effect by inhibiting the hypothalamic–pituitary–adrenal (HPA) axis via a GABAergic neuronal projection to the paraventricular nuclei of the hypothalamus, resulting in hyperpolarization and inhibition of the target neurons [26]. These findings support our own results, indicating a state of anxiety in rats exposed to OTA.

Other studies have also shown that OTA leads to a decrease in striatal dopamine levels, which affects this neurotransmitter [15]. The dopaminergic system is involved in various aspects of brain function, including locomotion [27]. Dopamine depletion affects locomotion, leading to dysfunction and hypoactivity [28]. Our results, showing locomotor disruption and hypoactivity in Wistar rats exposed to OTA at a dose of 289 µg/kg, are consistent with previous work [15,27].

The anxiety state was also assessed by the dark/light test, which is commonly used to assess anxiety in rodents and is based on rats’ natural preference for dark areas [29]. Our results showed that OTA-exposed rats spent more time in the light zone compared with the control group, reflecting their anxious state. These disturbances may be due to the effects of ochratoxin A on the central nervous system, as demonstrated by Bhat et al., 2018 [30], who showed the inhibition of monoamine metabolism, including serotonin, dopamine and norepinephrine, as well as an overall decrease in neurotransmitter levels, corroborating the findings of Rommelfanger and Weinshenker in 2007 [31] of decreased levels of neurotransmitters such as serotonin and norepinephrine.

The intoxicated rats were treated with *Syzygium aromaticum* essential oil (EOC), which was injected intraperitoneally. They showed an improvement in their psychological state. This was reflected in increases in the numbers of tiles crossed and visits to the center, and even a reduction in stereotyped behaviors reflecting a state of anxiety. In parallel, the activity of eugenol has been compared with propofol at sedative levels, assessing the anti-anxiety effect on fish [32]. In addition, the work of Sahin et al., 2017 [33], shows that a *Syzygium aromaticum* extract significantly and specifically potentiates GABA (γ-amino butyric acid)-induced currents through an allosteric mechanism in a concentration-dependent manner (0.5–5 µg/mL; up to 426 ± 23%), and GABA implicated in the potentiation of GABAergic synaptic transmission is used as a well-established target for the treatment of insomnia, anxiety disorders, and epilepsy. Furthermore, the administration of Giroflier plant essence nano-encapsulated in β-cyclo dextrin decreases the stress response [34]. Consequently, eugenol is considered a neuroprotective therapy for stress-related diseases [35].

The forced swimming test (FST) is widely implicated in the evaluation of antidepressant drugs in rodents [36]. The results showed that OTA-intoxicated rats exhibited a significant increase in immobility time, which reflects a state of desperation; this means that the animal loses hope of escaping the stressful environment and we can interpret this as a state of depression and inability to maintain efforts to save themselves. The control rats, on the other hand, showed flight behavior and increased mobility time.

This behavior may be due to disturbances in the central nervous system, and as already mentioned, ochratoxin A causes an alteration and disruption of neurotransmitter levels, some of which are implicated in the development of depression, such as serotonin and dopamine. Dopamine stimulates serotonin production, while serotonin has an inhibitory effect on dopamine production. This explains the reduction in serotonin levels after administration of neurotoxins that impair dopamine secretion [37].

In fact, OTA accumulates in several areas of the brain, including the hippocampus, cortex, gray matter, and striatum, which contain a large number of dopamine receptors. The toxin’s neurotoxic effect results in dopamine depletion, which is implicated in Parkinsonism [38] and low serotonin stimulus, resulting in depression.

In addition, treatment with EOC showed an antidepressant effect, with the treated rats showing an improvement in their psychic state through the FST by a reduced immobility time, reflecting the state of flight from danger. These results are in line with Liu et al., 2015 [39], who, following the administration of *Syzygium aromaticum* EO to rats at a dose of 200 mg/kg by gavage, noted an antidepressant effect assessed in two tests, forced swimming and tail suspension, which gave significant results. Furthermore, the same study confirmed that long-term treatment with *Syzygium aromaticum* essential oil by gavage possesses an effective antidepressant property that significantly enhances the hippocampal pathway by improving levels of hippocampal extracellular signal-regulated kinase (p-ERK) proteins, cyclic-AMP response element-binding protein (p-CREB), and brain-derived neurotrophic factor in mice exposed to unpredictable chronic mild stress. In addition, according to Garabadu et al., 2011 [40], eugenol enhanced serotonin (5-HT) levels in all CNS regions, with reduced noradrenaline levels in all CNS regions except the hippocampus. Eugenol exerted its action by regulating the brain’s monoaminergic and hypothalamic–pituitary–adrenal systems.

The Morris maze was designed to assess spatial memory and learning in rats. In this test, OTA-intoxicated rats showed a significantly higher latency during the learning period than the control group, with poor memorization of the platform location and a decrease in the number of visits to the northwestern (NW) quadrant. These reflect impaired learning and memory in OTA-intoxicated rats.

Ochratoxin A accumulates in different brain regions, including the hippocampus, causing oxidative stress, which can induce the development of neurodegeneration through inhibition of protein synthesis and its apoptotic effects. Studies have also shown that OTA induces apoptosis in the locus niger, striatum, hippocampus, and other brain regions [38,41].

Delibas et al., 2003 [42], reported that OTA reduces the levels of NMDA receptors in the hippocampus, affecting cognitive function. Since it is already known that the hippocampus is the main structure involved in memory, learning, cognitive functions, and emotions, any alteration or injury to its structure can lead to memory impairment, Alzheimer’s disease, and other problems.

Furthermore, GABAergic transmission is essential for proper brain formation and function and regulates adult neurogenesis in the subgranular zone (SGZ) of the hippocampal dentate gyrus [43]. The hippocampus is one of the few regions of the adult brain, including the ventricular zone (in rodents), and as recently shown in the striatum in humans, that continues to produce new neurons [44]. Adult hippocampal neurogenesis is implicated in memory and learning, and disrupted neurogenesis is implicated in cognitive and mood disorders, including anxiety and depression [45].

In addition, treatment with the essential oil of *Syzygium aromaticum* significantly improved learning and memory compared to intoxicated rats, and the rats showed good memorization instead of the platform, reflecting a corrective effect of the *Syzygium aromaticum* essential oil on OTA-induced damage to the hippocampus [41].

In the same context, the study by Adli et al., 2014 [46], shows that the administration of *Syzygium aromaticum* in Alzheimer’s disease mice produces improvements in cognition and behavior. This was confirmed by swimming exercise, which significantly increased BDNF levels in the hippocampus of treated mice. Also, a study by Parvizi et al., 2022 [47], shows that clove oil consumption may be a good therapeutic strategy to prevent the progression of Alzheimer’s disease by reducing apoptosis and improving mitochondrial homeostasis; clove oil also has cholinergic and glutaminergic effects [48].

Akbar et al., 2021 [49], reports that eugenol increased the numbers of putative neural stem cells (NPCs) and granule cells (GCs), and decreased neuronal cell death in the dentate gyrus DG, along with an increase in the dendritic complexity of neurons in the dentate gyrus (DG) region; whereas in ammonis horn 1 (CA1), eugenol only had a positive effect on the basal surface.

Similarly, Akbar et al., 2021 [49], reported that eugenol has neuroprotective effects, including on memory and learning, when tested on mice using the Y maze test and the Morris water maze, where eugenol increased and improved spatial memory and recognition in mice.

The test showed a significant increase in blood glucose levels in samples from the OTA-intoxicated group compared with controls. This test was carried out as a precursor to psychological stress. This is because, in a state of stress, blood glucose levels and blood pressure rise under the effect of adrenaline, the aim of which is to ensure the transport of glucose and oxygen to all the body’s organs. Our results are in agreement with [50], which noted an increase in glucose levels following administration of 1 μM OTA. Hyperglycemia is a risk factor for neuronal dysfunction, particularly altered signaling mechanisms [51].

After the administration of *Syzygium aromaticum* EO, the results show a decrease in blood glucose levels in intoxicated rats treated compared with intoxicated rats, and so these results rest on the fact that the EO reduces the stress response. Several results are in agreement with our findings. The study by de Oliveira et al., 2019 [52], shows that eugenol administration results in a decrease in blood glucose levels [52], and a dose of 50 mg/L eugenol is capable of reducing blood glucose levels [53]. Another study by Amanda in 2020 [34] reports that the administration of nano-encapsulated nail EO produces a decrease in blood glucose levels. In addition, work by Solomon et al., 2019 [54], in diabetic rats found that eugenol extracted from HEC had an anti-diabetic effect, regulating blood glucose levels through the inhibition of glycogen phospholipase and stimulation of glycogen synthase.

Our results showed a significant increase in the level of cortisol in the blood of the rodents compared to the control group. We measured the level of cortisol in the blood because it is a biomarker of psychological stress. In a stressful situation, the amygdala, which is involved in the evaluation of behavioral responses related to fear and anxiety, sends a stress signal to the hypothalamus. This activates the sympathetic nervous system (SNS) and the adrenal glands, leading to increases in the levels of catecholamines such as epinephrine, which increase heart rate, while activating the hypothalamic–pituitary–adrenal (HPA) axis. Initially, the hypothalamus releases corticoliberin (CRH), which stimulates the pituitary gland to release adrenocorticotropic hormone (ACTH). Once in the bloodstream, ACTH reaches the adrenal glands above the kidneys, where it binds to receptors in the fasciculated zone of the adrenal cortex, triggering the production of cortisol. Our results corroborate the observations of Kumar et al., 2013 [55], who also observed an OTA-induced hormonal imbalance, particularly of cortisol.

Moreover, after administration of clove EO, the biochemical result for cortisol revealed an observable reduction in blood cortisol levels in intoxicated rats treated with this oil. Our results are in agreement with the study by de Oliveira et al., 2019 [52], which shows that administration of eugenol results in a reduction in blood cortisol levels compared with control fish.

The histological study revealed that acute exposure to OTA caused lesions and hemorrhages in diverse brain regions, as well as signs of degeneration, including eosinophilic Purkinje neurons. These results are in agreement with those of [30], which observed that OTA causes various brain lesions, such as meningeal, cortical, and subependymal hemorrhages, and stratification changes in the cerebellum. In addition, we observed gliosis, a process of glial cell proliferation in response to brain damage [56], indicating ochratoxin A-induced disturbances in the brain.

The damage caused by OTA in the brain tissue is linked to its impact on oxidative stress (lipid peroxidation and SOD activity), affecting several brain regions and increasing its toxicity [38,57]. Its genotoxic effect and impact on the cell cycle cause cellular dysfunction and imbalance and disrupt cell adhesion by interfering with proteins. The accumulation of OTA in different areas of the brain, its apoptotic potential that alters cell signaling, and the oxidative stress it causes, which generates free radicals capable of altering cellular functions, are among the main mechanisms of ochratoxin A toxicity [57]. Ochratoxin A is also known to affect DNA, disrupting protein synthesis and leading to a loss of cellular homeostasis and tissue damage [17].

The use of *Syzygium aromaticum* essential oil was shown to have a beneficial effect on the brains of rats intoxicated by OTA. Rich in polyphenolic compounds, tannins, and flavonoids, this essential oil has a powerful antioxidant effect, protecting the brain structure. It also acts as a neuromodulator by inhibiting acetylcholinesterase (AChE) activity, thereby promoting better signaling and brain activity [58].

The pharmacological results obtained, particularly the improvements in memory and cognitive function of rats treated with clove essential oil, indicate a promising neuroprotective potential. However, to better understand the mechanisms underlying these effects, it is essential to explore the molecular interactions of active compounds, such as eugenol, with specific targets in the brain. In this respect, molecular docking was used as a complementary method to identify receptors and enzymes likely to be modulated by the essential oil. In our study, we performed molecular docking between two molecules, ochratoxin A and eugenol, to study their respective affinities for the hippocampal NMDA receptor. The structures and data for these molecules were extracted from the PubChem and Swiss-Prot databases.

Using the AutoDock program, we were able to simulate the formation of these complexes and evaluate their affinities. The results were then analyzed using PyMol and LigPlot+ [59]. According to the results, two molecules have affinity with the NMDA receptor, and the best ligand conformations were analyzed for their binding interactions by the binding free energies (docking affinity, kcal/mol). The results are shown in Table 3 and Figure 10 and Figure 11.

According to the table and among these proteins, ochratoxin A had the best affinity with the NMDA receptor, and its interaction with the receptor was a hydrogen-bonding interaction at amino acid Ile 548 with a distance of 2.89 Å. OTA has already been reported to have an effect on NMDA receptors by reducing their number in the hippocampus, as reported by [42,60,61].

This was followed by eugenol, with a hydrogen bond interaction at the Lys 458 amino acid with a distance of 2.84 Å. According to the literature, eugenol has an inhibitory action on NMDA receptors to induce anesthetic and analgesic effects [62], which shows that binding is possible between eugenol and the NMDA receptor, as reflected by the result of the docking performed as shown in the following Figure 11.

## 4. Materials and Methods

### 4.1. Extraction and Characterization of EOC by GC/MS

The flower buds of the clove plant (*Syzygium aromaticum*), which is native to the island of Molucca in Indonesia, are widely used in Algeria, both for their importance in traditional cooking and for their medicinal properties. Available all year round on the local market, dried, ripe *Syzygium aromaticum* seeds were purchased at the Saïda marketplace (Algeria). They were identified and authenticated by Professor Hasnaoui, an expert in taxonomy. A reference specimen, coded P-200676, has been preserved in the herbarium of the Biology Department of the Dr. Moulay Tahar University in Saïda.

Extraction of the EO was carried out using a Clevenger-type hydrodistiller. The protocol for this method involves immersing 20 g of crushed cloves (the part of the plant used are the flower buds) in 250 mL of distilled water in a heat-resistant flask at a moderate temperature, which is successively raised to the boiling point by a flask heater until approximately 80 mL of distillate is collected. At the end of the experiment, the collected liquid was placed in a decanted ampoule to separate the two phases (aqueous and organic) to obtain the essential oil [63]. An analysis of this essential oil was performed using GC/MS on a Varian Chrompack-CP 3900, with 0.2 μL of the non-polar extract injected. Helium (He) was used as the carrier gas at a flow rate of 0.3 mL/min. The capillary column used was a VF5 (stationary phase: 5% phenyl-polysinoxane and 95% methyl), measuring 30 m long and a 0.25 mm internal diameter, with a phase thickness of 0.25 μm. The temperature of the injection column started at 70 °C, was maintained for 2.5 min, then increased in steps of 15 °C/min to 255 °C, and was maintained for 20 min. The analysis used a mass spectrometry detector (Saturn 20200) set at 250 °C. The system was controlled by specific software, with a NIST and PubChem compound identification database.

### 4.2. Animal Origin and Housing Conditions

In our study, the animal model used was the albino Wistar rat, weighing 200–300 g ± 50 g, kept in the animal house of the Laboratory of Biotoxicology, Pharmacognosy and Biological Valorisation of Plants (LBPVBP) of the Biological Science Department at the Université Dr Moulay Tahar, in Saïda, Algeria. The animals were housed in plastic cages (43 × 28 × 15 cm) in a controlled environment with a light cycle of 12 h of darkness and light, at a temperature maintained between 22 and 23 °C. They had permanent access to bottled water and food. The number of animals was kept to a minimum, in line with European Council directives (86/609/EEC).

#### Dividing the Groups

First, we divided the 32 rats into 4 groups according to body weight: the control group contains 8 rats, with body weights of 200 and 300 g ± 50 that were given food and water only; the OTA intoxicated group contains 8 rats receiving an OTA solution intraperitoneally at a dose of 289 µg/kg/w.b. solubilized in 1 M NAHCO_3_ every 48 h for 12 doses [64]; the intoxicated and treated (OTA-EOC) group contains 8 OTA-intoxicated rats treated with clove EO at a dose of 0.1 mL/Kg/bw diluted in 40 µL of physiological water with TWIN 80 drops [46]; and a control group treated with *Syzygium aromaticum* EO that contains 8 rats receiving clove EO at a dose of 0.1 mL/Kg/bw diluted in 40 µL of physiological water with drops of TWIN 80 [46]. Body weight was assessed by weighing the rats daily throughout the experimental period, and the brain weight of Sinc batches was recorded after sacrifice for the evaluation of body weight and brain weight.

The number and suffering of animals were minimized in accordance with the guidelines of the European Council Directive (86/609/EEC).

### 4.3. Neurobehavioral Tests

#### 4.3.1. Forced Swimming Test

This experiment evaluated a depressive behavior, the state of resignation, observed in rodents, characterized by increased immobility, as described by its initiator [65]. The experimental apparatus consists of a transparent Plexiglas cylinder measuring 20.7 cm and 39 cm (diameter and height), three-quarters filled with water at 21 °C. A 6 min forced swimming test was performed on the rats. The parameters measured were as follows: mobility time “when the animal is actively swimming with all four legs” and immobility time “when the animal floats with weak movements, reflecting a state of behavioral despair”.

#### 4.3.2. Open Field Test

This test is used to evaluate spontaneous locomotion, the anxiety state and discovery of a new environment [66]. This test is a large open rectangular box with a width of 75 cm/75 cm and height of 35 cm, edged on the floor by 10 cm tiles. The rat is placed in 1 of the 4 corners of the open field frame, their head is exposed to corners (wall), and their behavior is assessed for 6 min. The six parameters evaluated in this test are the latency time (in seconds), corresponding to the time it takes the rat to emerge from the four corner tiles; the total number of tiles crossed by the rat during the test (6 min), reflecting its locomotor activity; number of times the rat visits the central nine tiles; number of sit-ups; number of grooming behaviors; and number of defecations. This test assesses the rat’s capacity for exploration in a stressful environment. The number of tiles crossed and the number of sit-ups indicate its exploratory activity and its emotional state [67].

#### 4.3.3. The Morris Water Maze

This test, designed to evaluate spatial memory and learning in animal models, is a large cylinder with a 136 cm and 60 cm diameter and height, respectively, and a circular platform (10 cm diameter and 28 cm height) placed in the northwest quadrant. The pool was filled with water (23 °C ± 2 °C) mixed with a white material until the platform was fully immersed at 1 cm from the water surface. In the experimental room, visual cues were placed at different points around the pool, and the rats were exposed for 4 days of training. Each training day consisted of a block of four trials (north, west, south, and east), each trial lasting 60 s, and between each trial 45 min to 1 h was allowed for each rat to rest. The rat was placed in the target quadrant with its head facing the walls and swam freely for 60 s until it found the hidden platform; if it had not found it, it was placed on the PLT for about 20 s, and so on until all four trials were complete. Day 5 had two tests: the probe test (exit from the hidden platform) was carried out 24 h after the end of training (day 5) by measuring the time spent in the target quadrant over 90 s, and the visual test to assess the state of vision by attaching a light-colored flag to the platform and recording the results [68]. This test assessed not only their ability to memorize and learn but also their problem-solving strategy (probe test and hidden PLT).

#### 4.3.4. The Dark/Light Test

This test is made up of four chambers, two of which are dark black, and the other two are illuminated in white, in order to create an ambient environment for the rat. The rat was placed in the middle of these four chambers with free access that allowed movement on both sides, and the test was performed within 30 min, knowing that rats are nocturnal animals that prefer dark places. In these 30 min, a video camera recorded their movements and time spent in each compartment (limuneux/dark). If the recorded time spent in the lit compartment was higher, this may represent a behavioral indicator of anxiety; under normal conditions, rats record a very low time in the lit compartment [69].

##### The Progress of the Tests

We performed the forced swimming test with an open field test in the same day spaced by two hours. In addition, the dark and light test was performed in one day alone. However, the Morris water maze test was conducted during 5 days of testing alone because the trial number that exists in this test can influence the behavior of the rats.

### 4.4. Biochemical Assays

#### 4.4.1. Blood Glucose Levels

The method used for the blood glucose determination is colorimetric (SPINREACT Kit). Glucose oxidase (GOD) catalyzes the oxidation of blood glucose to gluconic acid, forming hydrogen peroxide (H_2_O_2_). The latter is detected by phenol-aminophenazone in the presence of peroxidase (POD). The intensity of the color formed is proportional to the concentration of glucose in the sample, calculated by a spectrophotometer at a wavelength of 505 nm. In this study, blood samples were taken from fasting rats after neurobehavioral tests, ensuring standardized measurements [46].

#### 4.4.2. Cortisol Levels

Blood cortisol levels were mainly determined routinely using immunological techniques. The analytical detection limit is generally 1.00 nmol/L or 0.36 g/L, based on the binding of cortisol to a specific antibody. In the test tube, there is binding competition between the antibodies and any carrier proteins for cortisol [70].

### 4.5. The Brain Histology Study

Samples (brain tissue) were taken from each group of rats after sacrifice, fixed with 10% buffered formalin, dehydrated in a series of ethanol baths of increasing concentrations (from 70% to 100%), then clarified with xylene, and embedded in paraffin. The resulting blocks were cut into 5 μm-thick sections, systematically stained with hematoxylin and eosin (H&E) using the method of Bancroft (1975), and then observed under a microscope at ×40 magnification.

### 4.6. In Silico Docking Study

The in silico study proceeded for two volatile components (eugenol and eugenol acetate), which were phytochemically identified from *Syzygium aromaticum* (clove), with one neurotoxic compound (ochratoxin A (OTA)) and four selected neurotransmitters/controls (dopamine, glutamic acid, norepinephrine, and serotonin); their SMILES were all downloaded and the protocol was validated as mentioned in [71,72]. Two explored neurotargets, including human acetylcholinesterase receptor (PDB ID: 4EY5, 2.30 Å) [73,74] and human butyrylcholinesterase receptor (PDB ID: 6I0C, 2.76 Å) [74], were obtained from PDB (Protein Data Bank, its link is https://www.rcsb.org (accessed on 11 June 2024 and 4 July 2024). The molecular docking study was performed using the program “PyRx Autodock Vina” (Scripps Research, La Jolla, CA, USA), as previously mentioned in [71,72,75].

### 4.7. Statistical Analysis

The data are presented in the form of the means (Ms) of the individual values, together with the standard errors of the means (S.E.Ms.). Two means were compared using Student’s *t*-test. To compare several means, an analysis of variance (ANOVA) was performed by including the intoxication factor (Ochra A, T) and/or the treatment factor (EO, ED), with a Student–Newman–Keuls post hoc test where appropriate. Repeated measures ANOVAs were used to analyze the time factor. A *p*-value < 0.05 was considered statistically significant. Statistical analyses were performed using SigmaStat software 3.5 (SPSS Inc., Chicago, IL, USA).

## 5. Conclusions

Our research has demonstrated that *Syzygium aromaticum* essential oil exerts a beneficial effect against acute ochratoxin A intoxication in Wistar rats, as manifested by improvements in various neurobehavioral tests (antidepressant (positive learning and memory) and anti-anxiety) and the correction of biochemical parameters (blood glucose and cortisol levels). Regarding the in silico results for two cholinergic proteins, they supported the obtained in vivo results, wherein eugenol acetate existed in the *S. aromaticum* (clove) when compared to eugenol in the volatile oil and the neurotransmitter controls, and it seems that the highest potent neuroprotective agents depended on the monoterpene subclass present in *S. aromaticum*, as revealed in the examined enzymes. The ability of the substances to inhibit esterase enzymes was demonstrated by a continuous analysis and in vitro and in vivo studies that are well-executed and illustrated in this manuscript to configure the stability and substrate performance of the molecular interactions for the inhibitor(s), which are shown in Table 3 and Figure 10 and Figure 11. This suggests that the substances may be useful as first treatments for brain dysfunction and that they may be useful against pathogenic targets. Consequently, surprising findings revealed that all evaluated clove oils were more effective AChE inhibitors than BuChE inhibitors.

## Figures and Tables

**Figure 1 plants-14-00130-f001:**
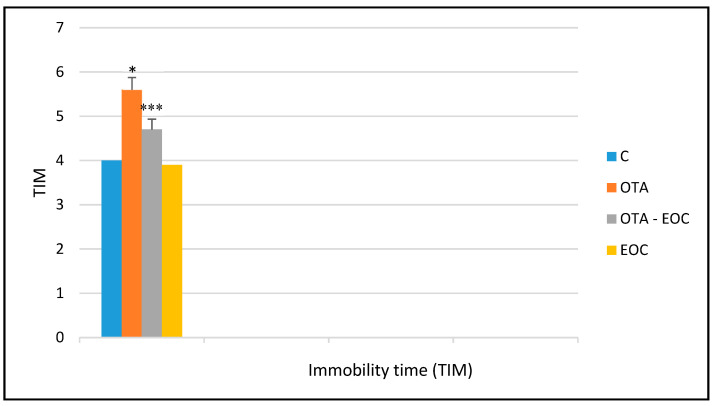
Forced swimming immobility times of control, OTA-intoxicated, and intoxicated and EOC-treated young animals (rats). The data are expressed as means ±SEMs (*** *p* < 0.001, * *p* < 0.05). Eight rats were included in each group.

**Figure 2 plants-14-00130-f002:**
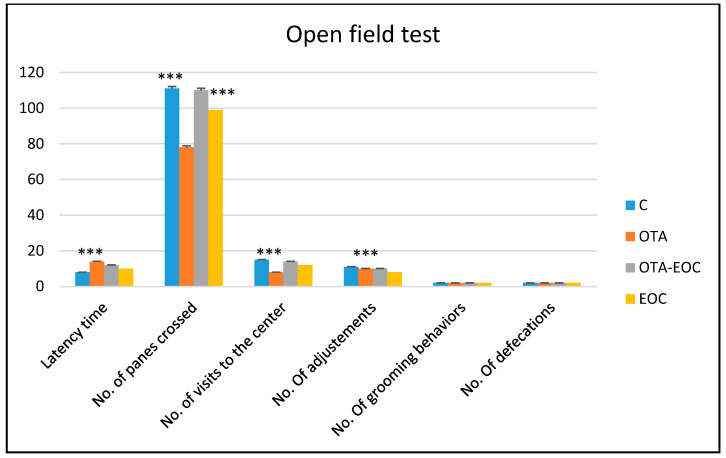
A comparison of the various parameters of the open field test in control, OTA-intoxicated, OTA-intoxicated and EOC-treated rats. The data are expressed as means ± SEMs; OTA vs. C (*** *p* < 0.001); OTA-EOC vs. OTA-EOC (*** *p* < 0.001). Eight rats were included in each group.

**Figure 3 plants-14-00130-f003:**
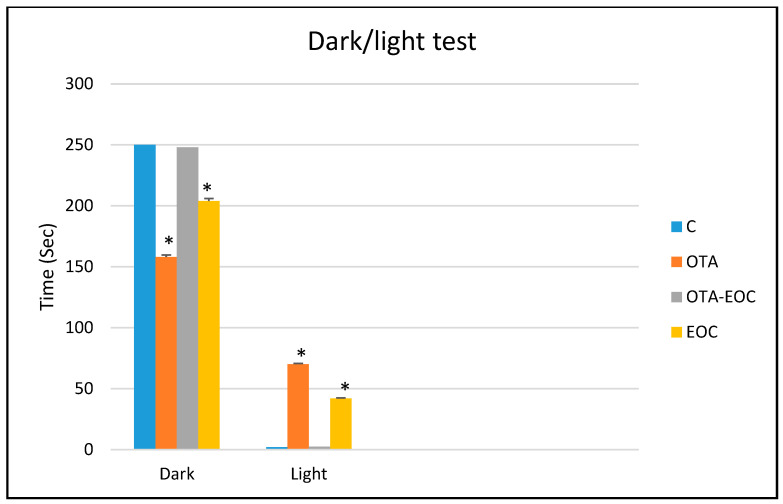
Time spent in the dark/light compartments by the four groups. The data are expressed as means ± SEMs; OTA vs. C (* *p* < 0.05), and OTA vs. OTA-EOC (*: *p* < 0.05). Eight rats were included in each group.

**Figure 4 plants-14-00130-f004:**
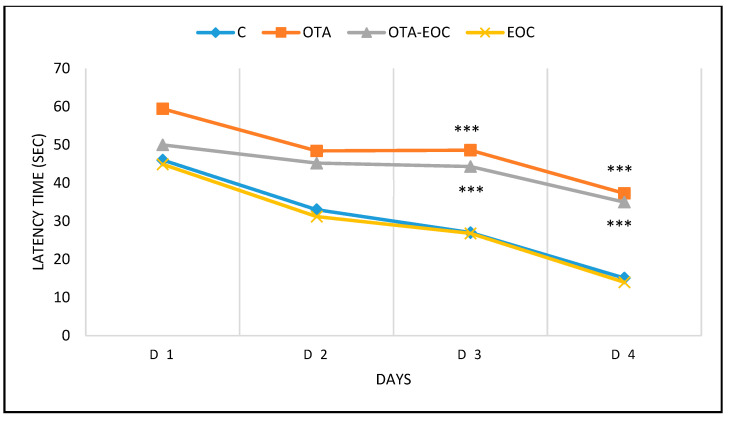
Morris water maze test: latency during the learning phase (over 4 days) of the control (C), treated (EOC), intoxicated (OTA) and intoxicated and treated (OTA-EOC) rats. The data are expressed as means ± SEMs (*** *p* < 0.001). Eight rats were included in each group.

**Figure 5 plants-14-00130-f005:**
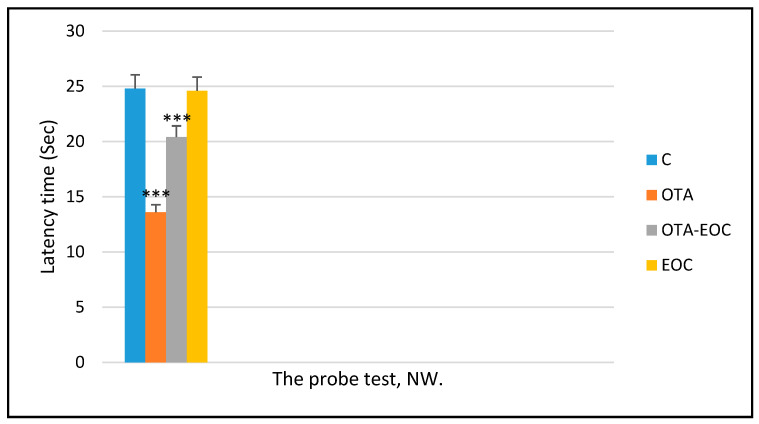
Time spent in the northwest quadrant (NW) during probe testing by control (C), treated (EOC), intoxicated (OTA), and intoxicated and treated (OTA-EOC) rats. The data are expressed as means ± SEMs (*** *p* < 0.001). Eight rats were included in each group.

**Figure 6 plants-14-00130-f006:**
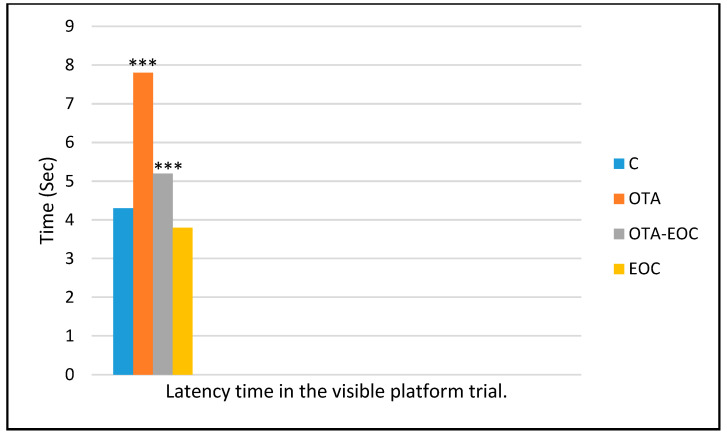
Latency during the visible platform trial of control (C), treated (EOC), intoxicated (OTA), and intoxicated and treated (OTA-EOC) rats. The data are expressed as means ± SEMs (*** *p* < 0.001). Eight rats were included in each group.

**Figure 7 plants-14-00130-f007:**
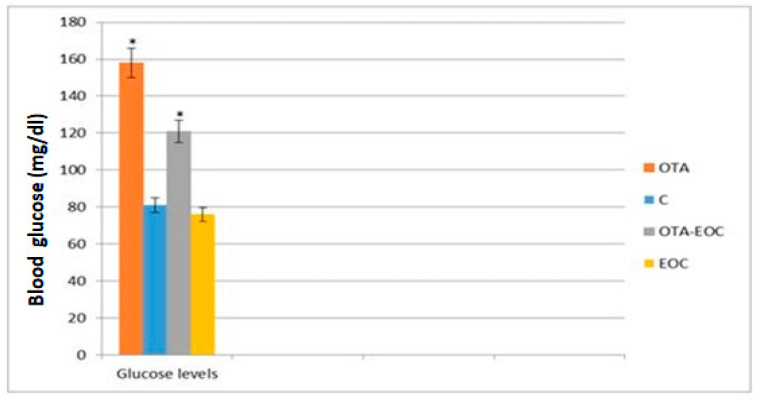
Blood glucose levels in the control, OTA-intoxicated, OTA/EOC-treated, and EOC-treated groups. The data are expressed as means ± SEMs; OTA vs. C (*: *p* < 0.05) and OTA vs. OTA-EOC (*: *p* < 0.05). Eight rats were included in each group.

**Figure 8 plants-14-00130-f008:**
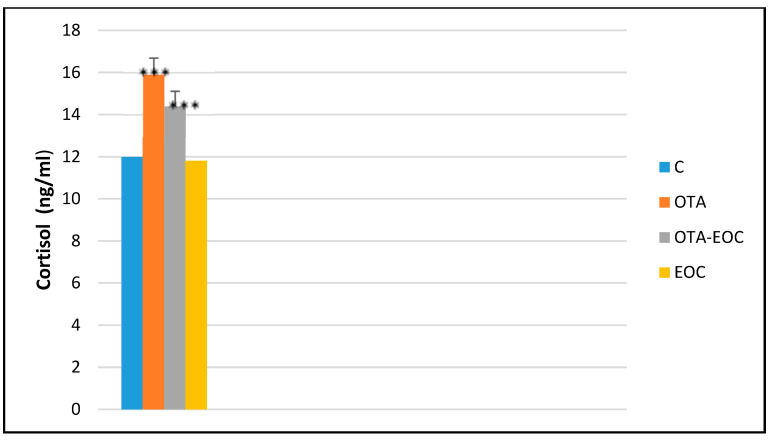
Blood cortisol levels in the control, OTA-intoxicated, OTA/EOC-treated, and EOC-treated groups. The data are expressed as means ± SEMs; OTA-EOC (*** *p* < 0.001). Eight rats were included in each group.

**Figure 9 plants-14-00130-f009:**
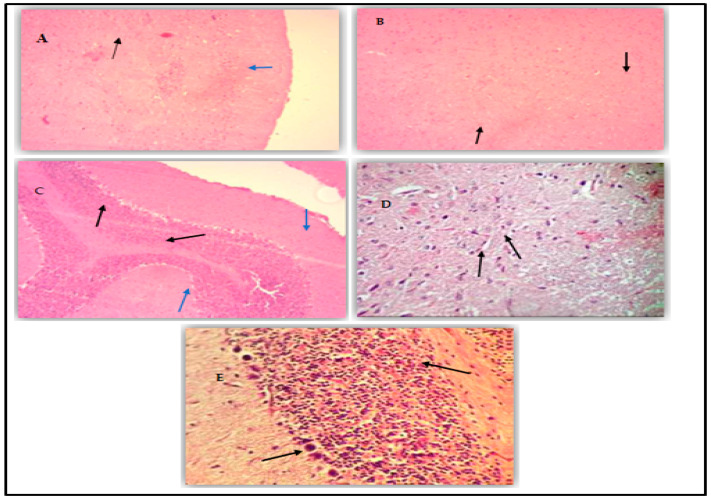
Histological sections of the rat brain. (**A**) Photomicrograph of a brain lesion in the hippocampus; pyramidal neurons are slightly eosinophilic (Black arrow) and a perivascular bulge is marked (Blue arrow). H&E staining, 100×. (**B**) Photomicrograph of a cerebral lesion, neurodegenerative changes, and gliosis. H&E staining, 100×. (**C**) Photomicrograph of a brain lesion with eosinophilic Purkinje neurons (Black arrow), marked congestion and hemorrhages in the brain and meninges (Blue arrow). H&E staining, 100×. (**D**) Photonic microscopic view of tissue from the cerebellar cortex stained with hematoxylin and eosin G (×40). Rats treated with EOC and intoxicated with OTA appeared to have a normal architecture. (**E**) Photonic microscopic view of tissue from the cerebellar cortex of a control rat, stained with hematoxylin and eosin G (×40).

**Figure 10 plants-14-00130-f010:**
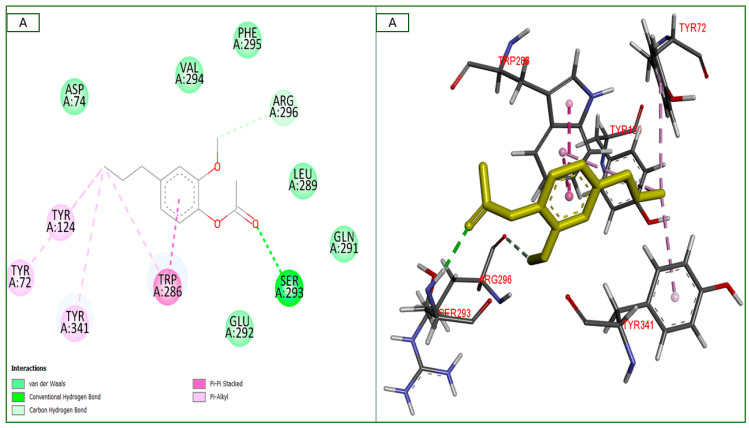
Two-dimensional and three-dimensional molecular interactions of the most prevalent essential compounds: (**A**) eugenol acetate; (**B**) eugenol identified in *Syzygium aromaticum* with the neurotransmitter “controls” (**C**) serotonin, (**D**) norepinephrine, and (**E**) dopamine; and the neurotoxic compound (**F**) ochratoxin A with the co-crystallized ligand “NAG” in the binding site of the human acetylcholinesterase receptor (PDB ID: 4EY5). Dimensions X, Y, and Z are 19.2037, 16.0972, and 19.7357, respectively, the root mean square deviation (RMSD) < 2, and components in 3D are highlighted in yellow.

**Figure 11 plants-14-00130-f011:**
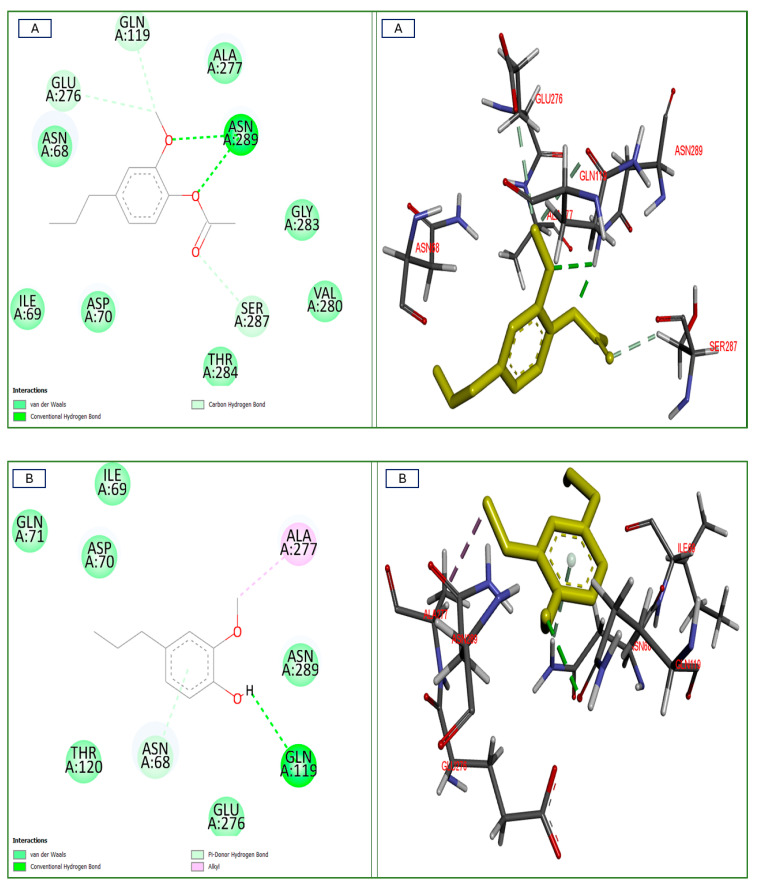
Two-dimensional and three-dimensional molecular interactions of the most prevalent essential compounds: (**A**) eugenol acetate; (**B**) eugenol identified in *Syzygium aromaticum* with the neurotransmitter “controls” (**C**) serotonin, (**D**) norepinephrine, and (**E**) glutamic acid; and the neurotoxic compound (**F**) ochratoxin A with the co-crystallized ligands “NAG, MES, and GZ5” at the docking site of the human butyrylcholinesterase receptor (PDB ID: 6I0C)/The dimensions X, Y, and Z are 18.0237, 19.1638, and 17.9918, respectively, the root mean square deviation (RMSD) < 2, and components in 3D are highlighted in yellow.

**Table 1 plants-14-00130-t001:** Gas chromatographic concentration and retention time of the compounds identified in the essential oil of *Syzygium aromaticum*.

Compounds	Retention Time (min)	Concentration (%)	Chemical Formula
Aromadendrene	54.2	0.01	C_15_H_24_
Calamenene	67.6	0.01	C_15_H_22_
*β*-Caryophyllene	53.4	7.21	C_15_H_24_
Caryophyllene Oxide	76.7	0.08	C_15_H_24_O
Chavicol	92.7	0.09	C_9_H_10_O
*α*-Copaene	46.0	0.03	C_15_H_24_
Cubenol	80.6	0.01	C_15_H_26_O
*α*-Cubebene	43.4	0.02	C_15_H_24_
Epoxide isomeric	73.9	0.01	C_28_H_30_O_2_
Epoxy-6,7-humulen	79.7	0.01	C_15_H_24_
Eugenol	85.3	80.83	C_10_H_12_O_2_
Eugenyl acetate	89.6	10.48	C_12_H_14_O_3_
*α*-Farnesene	61.9	0.02	C_15_H_24_
Geraniol	67.5	0.03	C_10_H_18_O
Germacren	50.4	0.01	C_15_H_24_
*α*-Himachalene	58.8	0.02	C_15_H_24_
*β*-Himachallene	60.9	0.01	C_15_H_24_
*γ*-Himachallene	58.8	0.01	C_15_H_24_
*α*-Humulene	57.8	0.87	C_15_H_24_
Isocaryophyllene	51.6	0.02	C_15_H_24_
Isomeric selinadiene	61.2	0.01	C_15_H_24_
Methyleugenol	76.9	0.01	C_11_H_14_O_2_
Sesquiterpenol	79.9	0.01	C_15_H_24_O_2_
Sequiterponic epoxide	72.8	0.01	C_28_H_30_O_2_
Zonarene	57.0	0.02	C_15_H_24_
**Groups in the constitution (%)**			
Phenols	80.93		
Sesquiterpene hydrocarbons	8.56		
Esters	10.48		
Terpenic alcohols	0.03		
**Total identified (%)**	**100**		

The retention index was calculated from the retention times relative to that of the n-alkane series, %: abundance percentage.

**Table 2 plants-14-00130-t002:** Evaluation of mean weight parameter values for control, OTA-intoxicated, OTA–EOC-treated, and non-intoxicated treated (EOC) rats.

The Group of Rats	C	OTA	OTA-EOC	EOC
Body weight	385 g (±0.48)	245 g (±0.048)	315 g (±1.091)	400 g (±0.48)
Weight of the brain	2.19 g (±0.0018)	1.47 g (±0.05)	1.92 g ***	2.19 g *** (±0.004)

Values are expressed as means ± SEMs (*** *p* < 0.001).

**Table 3 plants-14-00130-t003:** Docking analysis of the volatile compounds identified in clove plant against neuro-enzymes (binding affinities and type of interactions produced from the optimal conformations of each essential oil component into the proteins).

Proteins	4EY5			6I0C		
Ligands	BA	BN	AA residues	BA	BN	AA residues
Co-crystallized ligands						
NAG	−4.4	2	Arg296, Gln291, Glu292, Leu76, Leu289, Ser293, Trp286, Tyr72, Tyr341, Val294	−5.5	2 *	Ala277, Asn68, Asn289, Asp70, Gln119, Glu276, Gly283, Ser287, Thr120, Thr284, Val280
MES	--	--		−4.9	3 *	Ala277, Asn68, Asn289, Gln119, Glu276, Gly283, Ser287, Thr284, Val280
GZ5	--	--		−2.8	6	Ala277, Asn68, Asn83, Asn289, Asp70, Gln67, Gln71, Gln119, Gly115, Gly116, Gly121, Ile69, Leu125, Pro285, Thr120, Thr122, Thr284, Trp82, Tyr128, Tyr332
Essential oil compounds						
Eugenol	−5.9	6 *	Gln291, Glu292, Leu289, Phe295, Ser293, Trp286, Tyr72, Tyr341, Val294	−4.7	4	Ala277, Asn68, Asn289, Asp70, Gln71, Gln119, Glu276, Ile69, Thr120
Eugenol acetate	−6.6	5	Arg296, Asp74, Gln291, Glu292, Leu289, Phe295, Ser293, Trp286, Tyr72, Tyr124, Tyr341, Val294	−5.2	3	Ala277, Asn68, Asn289, Asp70, Gln119, Glu276, Gly283, Ile69, Ser287, Thr284, Val280
Neurotoxic substance						
Ochratoxin A	−4.2	9 *	Asp74, Glu292, Leu76, Leu289, Phe295, Ser125, Thr75, Trp286, Tyr124, Tyr341, Tyr337, Val294	−3.2	4 *	Ala277, Asn68, Asn289, Asp70, Glu276, Gly283, Gly333, Ile69, Pro285, Ser287, Thr120, Thr284, Tyr332, Val280
Controls						
Dopamine	−5.5	3 *	Arg296, Gln291, Glu292, Leu289, Phe295, Ser293, Trp286, Tyr72, Tyr341, Val294	−4.9	5	Ala277, Asn68, Asn289, Asp70, Gln119, Glu276, Gly149, Ile69, Thr120
Glutamic acid	−4.9	2	Arg296, Gln291, Glu292, Gly342, Leu289, Phe295, Phe297, Ser293, Trp286, Tyr341, Val294	−5.0	3	Asn68, Asn83, Asp70, Gln67, Gly116, Gly121, Ile69, Pro84, Thr120, Thr122
Norepinephrine	−5.5	5 *	Arg296, Leu76, Leu289, Phe295, Phe297, Ser293, Trp286, Tyr72, Tyr341, Val294	−5.2	3	Ala277, Asn68, Asn289, Asp70, Gln71, Gln119, Glu276, Ile69, Thr120
Serotonin	−6.0	3 *	Arg296, Asp74, Ser293, Trp286, Tyr72, Tyr124, Val294	−5.2	3 *	Ala277, Asn68, Asp70, Gln68, Gln119, Ile69, Phe290, Thr120, Thr284

BA = binding affinity (Kcal/mol); BN = no. of formed bonds; * = substrate for the unfavorable bonds from the original no. of formed bonds; AA residues = amino acid residues; NAG = 2-acetamido–2–deoxy–beta–D–glucopyranose; MES = 2– (n–morpholino)–ethanesulfonic acid; GZ5 = (2–{R–azanyl–{N}s–[6–[(6–chloranyl-1,2,3,4–tetrahydroacridin–9–yl)amino]hexyl]–3–(1–{H}–indol–3–yl)propanamide.

## Data Availability

Data are contained within the article.

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
