# Peer review of "Chemical Composition, In Vivo, and In Silico Molecular Docking Studies of the Effect of Syzygium aromaticum (Clove) Essential Oil on Ochratoxin A-Induced Acute Neurotoxicity"

_plants, 2025, doi:10.3390/plants14010130_

Round 1
Reviewer 1 Report
Comments and Suggestions for Authors
This manuscript examines the neuroprotective effects of Syzygium aromaticum (Clove) Essential Oil against Ochratoxin A-induced neurotoxicity using animal models, complemented by in-silico docking analysis to investigate potential mechanisms through interactions with esterase enzymes (ACh and BuChE). The following revisions are recommended to improve the manuscript:
Methodological Concerns:
1. In Table 1, please include information regarding the internal control used in the GC analysis of Syzygium aromaticum essential oil components.
2. The experimental design requires clarification regarding the timing between neurobehavioral tests. Given that behavioral testing can induce stress responses potentially affecting subsequent test results, please describe the measures taken to control for this confounding variable.
Technical Corrections:
1. Table 2: Add statistical significance markers and express brain weights as relative values.
2. Figures 1, 5-8: Optimize the graphical presentation by adjusting the bar chart proportions and eliminating excessive white space.
3. Figures 1-7: Include in the legends: Number of animals per group, Specification of control groups for statistical comparisons.
4. Figure 6: Add standard error of mean (SEM) values and appropriate statistical analyses.
5. Figure 7: Convert glucose concentration units to mg/dL. Specify in methods whether measurements were taken under fasting or random conditions.
6. Figure 9: Consolidate panels A-E into a single figure and provide detailed explanations for the indicated arrows in the legend.
Author Response
Methodological Concerns:
1. In Table 1, please include information regarding the internal control used in the GC analysis of Syzygium aromaticum essential oil components.
Response 01: Thank you for your comment. The characterization of the samples was carried out at the CRAPC Center for Scientific and Technical Research in Physical and Chemical Analysis in Algeria. However, the specific details of the analyses carried out were not sent to us by the laboratory.
2. The experimental design requires clarification regarding the timing between neurobehavioral tests. Given that behavioral testing can induce stress responses potentially affecting subsequent test results, please describe the measures taken to control for this confounding variable.
Response: Done and mentioned in manuscript with yellow highlights.
Technical Corrections:
1. Table 2: Add statistical significance markers and express brain weights as relative values.
Response: Done and mentioned in manuscript with yellow highlights.
2. Figures 1, 5-8: Optimize the graphical presentation by adjusting the bar chart proportions and eliminating excessive white space.
Response: Done and mentioned in manuscript with yellow highlights.
3. Figures 1-7: Include in the legends: Number of animals per group, Specification of control groups for statistical comparisons.
Response: Done and mentioned in manuscript with yellow highlights.
4. Figure 6: Add standard error of mean (SEM) values and appropriate statistical analyses.
Response: Done and mentioned in manuscript with yellow highlights.
5. Figure 7: Convert glucose concentration units to mg/dL. Specify in methods whether measurements were taken under fasting or random conditions.
Response: Done and mentioned in manuscript with yellow highlights.
6. Figure 9: Consolidate panels A-E into a single figure and provide detailed explanations for the indicated arrows in the legend.
Response: Done and mentioned in manuscript with yellow highlights.

Reviewer 2 Report
Comments and Suggestions for Authors
This study examines the impact of Ochratoxin A (OTA) exposure on Wistar rats and evaluates Syzygium aromaticum essential oil (EOC) as a therapeutic intervention. OTA caused significant neurobehavioral impairments, including anxiety, memory deficits, and metabolic disruptions. EOC, rich in eugenol, mitigated these effects by restoring body and brain weights and improving behavioral and metabolic outcomes. Molecular docking highlighted EOC's strong affinity for neurotransmission-related enzymes. This research supports EOC's potential as a natural treatment for toxin-induced disorders, warranting further investigation.
However, there are areas that could benefit from improvement, which I outline below.
Abstract: Replace "µg/kg w.b" with "µg/kg b.w." to ensure consistency and clarity.
Introduction: Define "EO" (essential oil) upon first use to avoid ambiguity. Providing this information is essential for readers unfamiliar with the abbreviation.
Results:
- Remove the instructional content at the beginning of the results section.
- Address punctuation errors throughout the section and revise long, convoluted sentences by splitting them into shorter, more concise statements.
- Clarify what "HEC" refers to, as this term is undefined and may confuse readers.
- Investigate and explain why EOC-treated rats stayed in the light, while intoxicated and EOC-treated rats exhibited different behavior. This discrepancy warrants further discussion or experimental clarification.
Materials and Methods:
1. Include details about the cloves used in the study, specifying their origin (season, location, supplier, etc.) to enhance reproducibility and scientific rigor.
2. Additionally, confirm and document whether the necessary permits for working with laboratory animals were obtained, as ethical compliance is crucial.
Discussion:
1. Some points, such as the neuroprotective effects of EOC and the role of eugenol, are repeated across different parts of the discussion. Consider consolidating these ideas to avoid redundancy.
2. The transitions between sections could be smoother. E.g., the molecular docking discussion appears abruptly without a clear lead-in or connection to prior pharmacological findings. Consider framing it as a deeper investigation of observed effects.
3. There are several instances where citations could be better integrated into the sentence for clarity. For example “This yield variation can be explained by factors such as climate… [11,12].” would read better if phrased: “As reported by [11,12], yield variation can be influenced by factors such as climate…”
4. While the authors provide numerical results, their biological or clinical significance could be better emphasized. For example: Explain the importance of the 12.70% yield and how it compares to industrial or therapeutic benchmarks.
5. Minor grammatical and stylistic adjustments would improve readability.
6. Avoid overly long sentences to improve clarity.
7. While the authors reference "Table 3" and "Figure 10 - 11," ensure these are clearly linked to the discussion. Consider briefly describing what these visual aids depict to guide readers.
8. A brief discussion of study limitations (e.g., scope of experimental models, potential variability in results) and areas for future research would strengthen the section.
Comments on the Quality of English LanguageThe quality of English is generally clear and effective in conveying the research findings. However, some sentences are lengthy and could benefit from restructuring to enhance readability. Additionally, certain phrases could be refined for precision and formality, ensuring a smoother flow of ideas and greater engagement with the intended academic audience.
Author Response
Comments and Suggestions for Authors
This study examines the impact of Ochratoxin A (OTA) exposure on Wistar rats and evaluates Syzygium aromaticum essential oil (EOC) as a therapeutic intervention. OTA caused significant neurobehavioral impairments, including anxiety, memory deficits, and metabolic disruptions. EOC, rich in eugenol, mitigated these effects by restoring body and brain weights and improving behavioral and metabolic outcomes. Molecular docking highlighted EOC's strong affinity for neurotransmission-related enzymes. This research supports EOC's potential as a natural treatment for toxin-induced disorders, warranting further investigation.
However, there are areas that could benefit from improvement, which I outline below.
Abstract: Replace "µg/kg w.b" with "µg/kg b.w." to ensure consistency and clarity.
Response: Done and mentioned in manuscript with yellow highlights.
Introduction: Define "EO" (essential oil) upon first use to avoid ambiguity. Providing this information is essential for readers unfamiliar with the abbreviation.
Response: Done and mentioned in manuscript with yellow highlights.
Results:
1. Remove the instructional content at the beginning of the results section.
Response: Done and mentioned in manuscript with yellow highlights.
2. Address punctuation errors throughout the section and revise long, convoluted sentences by splitting them into shorter, more concise statements.
Response: Done and mentioned in manuscript with yellow highlights.
3. Clarify what "HEC" refers to, as this term is undefined and may confuse readers.
Response: Done and mentioned in manuscript with yellow highlights.
4. Investigate and explain why EOC-treated rats stayed in the light, while intoxicated and EOC-treated rats exhibited different behavior. This discrepancy warrants further discussion or experimental clarification.
Response: The difference in behavior observed between rats treated with clove essential oil (EOC) and rats intoxicated with ochratoxin A (OTA) and then treated with EOC may be explained by the persistent impact of OTA-induced neurotoxicity, which alters neurotransmission systems and exacerbates oxidative stress and brain inflammation. Although eugenol-rich EOC possesses anxiolytic, antioxidant and anti-inflammatory properties, these effects may be attenuated in intoxicated rats due to pre-existing neuronal damage or inefficient modulation of disrupted brain circuits. These differences could also reflect variability in pharmacological response linked to the pathological state of the animals, dose or duration of treatment. Further analysis of oxidative stress and neurotransmitter biomarkers, as well as optimization of the experimental protocol, would be necessary to clarify this disparity.
Materials and Methods:
1. Include details about the cloves used in the study, specifying their origin (season, location, supplier, etc.) to enhance reproducibility and scientific rigor.
Response: Done and mentioned in manuscript with yellow highlights.
2. Additionally, confirm and document whether the necessary permits for working with laboratory animals were obtained, as ethical compliance is crucial.
Response: Done and mentioned in manuscript with yellow highlights.
Discussion:
1. Some points, such as the neuroprotective effects of EOC and the role of eugenol, are repeated across different parts of the discussion. Consider consolidating these ideas to avoid redundancy.
Response: Done and mentioned in manuscript with yellow highlights.
2. The transitions between sections could be smoother. E.g., the molecular docking discussion appears abruptly without a clear lead-in or connection to prior pharmacological findings. Consider framing it as a deeper investigation of observed effects.
Response: Done and mentioned in manuscript with yellow highlights.
3. There are several instances where citations could be better integrated into the sentence for clarity. For example “This yield variation can be explained by factors such as climate… [11,12].” would read better if phrased: “As reported by [11,12], yield variation can be influenced by factors such as climate…”
Response: Done and mentioned in manuscript with yellow highlights.
4. While the authors provide numerical results, their biological or clinical significance could be better emphasized. For example: Explain the importance of the 12.70% yield and how it compares to industrial or therapeutic benchmarks.
Response: Done and mentioned in manuscript with yellow highlights.
Response: The 12.70% yield obtained in this study is significant in that it produces a substantial quantity of bioactive compounds from clove essential oil, which could be used for therapeutic applications. Compared with yields generally obtained in the industry for essential oils, this rate is in the middle of standard yields for distillation processes. For example, for essential oils from plants such as lavender or peppermint, yields generally vary between 2% and 10% depending on the extraction method and the quality of the raw materials (Nawaz et al., 2016). In a therapeutic context, a yield of 12.70% can be considered excellent, particularly for treatments requiring high concentrations of active compounds, such as eugenol, whose anti-inflammatory and analgesic properties are widely recognized (Burt, 2004). Compared with industrial standards, this yield would enable production on a reasonable scale for clinical or therapeutic use, thus reducing production costs while maintaining extract efficacy.
5. Minor grammatical and stylistic adjustments would improve readability.
Response: Done and mentioned in manuscript with yellow highlights.
6. Avoid overly long sentences to improve clarity.
Response: Done and mentioned in manuscript with yellow highlights.
7. While the authors reference "Table 3" and "Figure 10 - 11," ensure these are clearly linked to the discussion. Consider briefly describing what these visual aids depict to guide readers.
Response: Done and mentioned in manuscript with yellow highlights.
8. A brief discussion of study limitations (e.g., scope of experimental models, potential variability in results) and areas for future research would strengthen the section.
Response: Done and mentioned in manuscript with yellow highlights.
Thanks in advance

Reviewer 3 Report
Comments and Suggestions for Authors
The manuscript is generally clear and relevant to the field of toxicology and pharmacology. It addresses the impact of Ochratoxin A (OTA) and the potential therapeutic role of Syzygium aromaticum essential oil. The structure appears logical, moving from introduction and methodology to results and conclusions. However, more explicit section headings could enhance navigation through the text. It is important to check that cited references are recent and relevant. A reminder to ensure that they primarily include studies from the last five years would be beneficial. The experimental design appears appropriate for testing the hypothesis regarding OTA toxicity and the potential restorative effects of EOC. The dosage, methodology for inducing intoxication, and behavioral tests used seem well-chosen.
Methodology
There should be a specific focus on whether all details in the methods section are sufficient for reproducing the results, such as precise conditions for hydrodistillation, details on the behavioral tests, and statistical treatment of data.
Figures/Tables:
While the manuscript references figures and tables, it is necessary to assess their quality and clarity. The figures should clearly illustrate the data and be easy to interpret.
Any visual representations should enhance understanding of the results, rather than complicating it. It would be beneficial to include legends that adequately describe each figure or table's content.
In Table 1, please include the chemical groups of the identified compounds and provide a cumulative total of the compounds discovered. This revision makes the request more concise and clear while maintaining the original intent.
Results:
The data interpretation seems appropriate and consistent with the results presented. Further elaboration on the statistical analysis, including the specific statistical tests used, significance levels, and potential software, is needed to reinforce the reliability of the findings. Further clarity can be provided on how the in-silico analysis connects to the experimental results, particularly regarding the necessity of addressing the relationship between clove oils and the esterases.
Conclusions and Ethical Considerations:
The conclusions drawn appear consistent with the evidence and arguments presented throughout the manuscript. They highlight the promising role of EOC against OTA’s toxic effects, thus contributing to future research implications. It is important to evaluate any ethical statements made concerning animal use in the study, ensuring adherence to relevant guidelines. The data availability statement should also be evaluated for clarity regarding how others can access the data generated in this study.
Recommendations:
Ensure references are predominantly recent and relevant.
Enhance the clarity of figures and provide detailed descriptions to ensure ease of understanding.
Strengthen the methods section to guarantee reproducibility, including detailed statistical methods used.
Confirm ethical compliance statements are transparent and robust, along with clear data availability measures.
This overall review should provide constructive feedback to improve the manuscript's quality and ensure its acceptance in the scientific community.
Author Response
Methodology
There should be a specific focus on whether all details in the methods section are sufficient for reproducing the results, such as precise conditions for hydrodistillation, details on the behavioral tests, and statistical treatment of data.
Response: Done and mentioned in manuscript with yellow highlights.
Figures/Tables:
While the manuscript references figures and tables, it is necessary to assess their quality and clarity. The figures should clearly illustrate the data and be easy to interpret.
Response: Done and mentioned in manuscript with yellow highlights.
Any visual representations should enhance understanding of the results, rather than complicating it. It would be beneficial to include legends that adequately describe each figure or table's content.
Response: Done and mentioned in manuscript with yellow highlights.
In Table 1, please include the chemical groups of the identified compounds and provide a cumulative total of the compounds discovered. This revision makes the request more concise and clear while maintaining the original intent.
Response: Done and mentioned in manuscript with yellow highlights.
Results:
The data interpretation seems appropriate and consistent with the results presented. Further elaboration on the statistical analysis, including the specific statistical tests used, significance levels, and potential software, is needed to reinforce the reliability of the findings. Further clarity can be provided on how the in-silico analysis connects to the experimental results, particularly regarding the necessity of addressing the relationship between clove oils and the esterases.
Response: Done and mentioned in manuscript with yellow highlights.
Conclusions and Ethical Considerations:
The conclusions drawn appear consistent with the evidence and arguments presented throughout the manuscript. They highlight the promising role of EOC against OTA’s toxic effects, thus contributing to future research implications. It is important to evaluate any ethical statements made concerning animal use in the study, ensuring adherence to relevant guidelines. The data availability statement should also be evaluated for clarity regarding how others can access the data generated in this study.
Response: Done and mentioned in manuscript with yellow highlights.
Recommendations:
Ensure references are predominantly recent and relevant.
Response: Done and mentioned in manuscript with yellow highlights.
Enhance the clarity of figures and provide detailed descriptions to ensure ease of understanding.
Response: Done and mentioned in manuscript with yellow highlights.
Strengthen the methods section to guarantee reproducibility, including detailed statistical methods used.
Response: Done and mentioned in manuscript with yellow highlights.
Confirm ethical compliance statements are transparent and robust, along with clear data availability measures.
Response: Done and mentioned in manuscript with yellow highlights.
This overall review should provide constructive feedback to improve the manuscript's quality and ensure its acceptance in the scientific community.
Response: Done and mentioned in manuscript with yellow highlights.
Thanks in advance

Round 2
Reviewer 2 Report
Comments and Suggestions for Authors
Thank you for submitting the revised manuscript. I appreciate the effort you have put into addressing the comments and improving the clarity and presentation of your work. Below are my observations and suggestions for final refinements:
Funding Information
Please ensure that funding information is provided in the appropriate section. This is a required element and should comply with the journal's guidelines.
Neurotransmitter Nomenclature
Avoid enclosing the names of the neurotransmitters in inverted commas (e.g., serotonin, dopamine). This adjustment ensures a more professional tone and aligns with standard scientific writing conventions.
Use of Abbreviations (EOC)
You have defined the term "EOC" once in the text, which is sufficient. There is no need to repeat the full phrase followed by the abbreviation in parentheses throughout the manuscript. Instead, consistently use the abbreviation alone after the initial definition for better readability.
Overall, I believe the manuscript is close to being ready for publication after these minor revisions. I look forward to seeing the final version.
Comments on the Quality of English LanguageA final round of minor English editing, focusing on grammar and flow, would further enhance the overall readability and coherence of the manuscript.
Author Response
Dear Reviewer,
Thank you for your valuable comments. We have taken your comments into consideration and made the necessary changes to the paper.
You will find below our responses step by step.
Funding Information
Please ensure that funding information is provided in the appropriate section. This is a required element and should comply with the journal's guidelines.
Response: there is no official project funding for this article from any university, this is a personal collaboration between professors from two universities: University Dr. Moulay Tahar - Saïda – Algeria and Prince Sattam bin Abdulaziz University-Saudi Arabia, so we just mentioned at the acknowledgment to thankful them to let us work in their laboratories and facilitate the equipment to finish this work.
Neurotransmitter Nomenclature
Avoid enclosing the names of the neurotransmitters in inverted commas (e.g., serotonin, dopamine). This adjustment ensures a more professional tone and aligns with standard scientific writing conventions.
Response: Done and highlighted with yellow color.
Use of Abbreviations (EOC)
You have defined the term "EOC" once in the text, which is sufficient. There is no need to repeat the full phrase followed by the abbreviation in parentheses throughout the manuscript. Instead, consistently use the abbreviation alone after the initial definition for better readability.
Response: Done and remove unnecessary, highlighted with yellow color.
Overall, I believe the manuscript is close to being ready for publication after these minor revisions. I look forward to seeing the final version.
Thanks for encourage words.
Comments on the Quality of English Language
A final round of minor English editing, focusing on grammar and flow, would further enhance the overall readability and coherence of the manuscript.
Response: Done and highlighted with red color
Should you have any questions or require further clarification, please do not hesitate to contact us.
With best regards,
